# Research of Glamping Tourism Based on the Aesthetics of Atmosphere

Ting Sun [1] and Tai Huang [1,2,*]

1    Department of Tourism Management, School of Social Sciences, Soochow University, Suzhou 215123, China
2    Academy of Culture and Tourism Research, Soochow University, Suzhou 215123, China
*    Correspondence: huangtai_fx@163.com

**Abstract:** Glamping, with its pursuit of connection with nature and focus on refinement, inspires the dissemination of glamping scenes—A tourism phenomenon that has received little attention in the literature despite its popularity with travelers. This study views the aesthetics of atmosphere theory as a breakthrough, analyzing user-generated content and in-depth interviews to identify three dimensions—physical environment, situation, and context—that are complementary developments of the aesthetics of atmosphere theory. After clarifying the characteristics of the construction of glamping scenes, we further explore the path of sustainable development of glamping from online to offline communication through the Stimulus-Organism-Response (SOR) framework. The results of grounded theory are used as stimulus and organism factors in the communication path and travel experience-sharing behavior and word-of-mouth recommendations are used as response factors. Research has revealed that physical environments and situational interactions play a decisive role in contextual perception and that unique organismic perception helps campers to respond positively to shared communication. The exploration of communication paths integrates atmospheric aesthetics theory and the SOR framework, contributing to the development of theory as well as suggesting ideas and references for the sustainable communication of glamping practices.

**Keywords:** the aesthetics of atmosphere; glamping; SOR framework; tourist scene; RED app

## 1. Introduction

In contrast to the overall slow recovery of the tourism market due to the normalization of the COVID-19 pandemic, China's glamping market has exploded, with 2020 called the first year of glamping in China. Unlike the luxurious style of American camping the term can connote, glamping in China draws more from the delicate style of Japanese camping. Glamping combines the consumption habits and travel preferences of young travelers to form a special cultural phenomenon and has been promoted on internet platforms to become a leisure activity with a strong development trend. Glamping plays an important role in enhancing human well-being as a spatial vehicle linking natural landscapes and human economies and societies [1]. Despite glamping's popularity in popular culture, it has received minimal attention in the literature. As Brooker and Joppe note, "The emergence of glamping has created an area in need of academic research" [2]. This study uses the aesthetics of atmosphere theory to reveal the elements of ambiance in the glamping scene and uses the SOR model to focus on the communication development path of glamping from online to offline. His is achieved by using grounded theory and content analysis to identify the conceptual dimensions of glamping and adopting inductive analysis to sort out the relationship between the physical environment, situation, and context. Additionally, analyzing the unique image of glamping. Through this process, it integrates the atmospheric aesthetic dimensions identified in the glamping tourism perspective into the SOR theoretical framework, theoretically extending the atmospheric aesthetics theory. By addressing the question of what the glamping atmosphere scene is, it

continues to explore why glamping has gained widespread popularity and become a hot phenomenon at the moment. The path of communication between the online and offline development of glamping answers the question of the atmosphere as an intermediate quality acting between the scene and the tourist. Therefore, this study will clarify the response mechanisms between the various elements of glamping and provide a useful reference for the construction of campsites in practice.

## 2. Literature Review

### 2.1. Glamping

The term glamping originated from the words "glamorous" and "camping"; it is a form of upscale outdoor experience that blends wild luxury and camping, as travelers' desire for natural surroundings merges with a growing demand for comfort as well as luxury [2], an emerging concept in camping [3]. The term "glamping" is used in the existing literature to refer to outdoor tourism activities or outdoor hotels; studies have focused on customers' motivations for choosing luxury camping hotels [4] and particular service quality ratings [5] or remaining loyal to particular ones [3], though there are limited results from the research on glamping as an outdoor tourism activity [6]. Glamping differs from traditional camping activities (e.g., those that use rudimentary sleeping bags and basic equipment) by emphasizing comfortable and innovative accommodations such as yurts, conical tents, tree houses, mobile homes, "bubble hotels", and villas [7,8], as well as creating a romantic and lightly luxurious atmosphere. Blending outdoor aesthetics and lifestyle, glamping often entails a combination of camping equipment and natural scenery, providing travelers with a pro-nature escape from everyday life [9]. In this study, glamping as a nature-integrated tourism activity is considered a new way of perceiving and engaging with nature. The aim is to immerse oneself in the countryside, while at the same time enjoying high-quality facilities and related activities that enhance the experience [8].

Previous research has confirmed that engagement with nature can bring visitors a sense of relaxation, relief from anxiety, the rewards of freedom and fun, and a sense of "smallness" or self-release [10]. Against the backdrop of the raging COVID-19 pandemic, travelers now prefer having distance from others, visiting natural spaces, and engaging in outdoor recreation, which act as driving factors behind the explosion of glamping [11]. The findings of Craig and Karabas also suggest that the widespread availability and accessibility of glamping cause it to be a viable leisure travel option during or after the pandemic [12]. In China, glamping has taken its cues from European and American wilderness comfort, while its aesthetics are regulated by Japanese-style camping culture, forming a new generation of camping that emphasizes nature, equipment, and a sense of atmosphere. With the innovative development of new media technology, the openness of social media platforms and the sharing of travel experiences are to a certain extent compatible and mutually constructive with the experience of glamping itself; the proliferation of glamping is accelerated by the beautiful pictures and detailed tips shared on social media platforms. The inherent leisure and social attributes of glamping match the experiential marketing attributes of the Red app; there is a natural bond between the two that lends itself to research.

The current research on glamping is still somewhat flawed. First, there is a lack of consistency in terminology and connotation when examining the subject of glamping. The current research usually treats glamping as a sub-topic of hotel management research, equating campgrounds to wild luxury hotels. While it describes the facilities and equipment, service quality, and reputation of the hotels, it lacks attention on the activity of glamping itself. As a new form of tourism, glamping differs from traditional camping in obvious ways, but they lack systematic organization and elaboration in the literature. Second, despite scholars' recognition of the role of nature in glamping, there remains a lack of empirical exploration of the perceived dimensions of the glamping experience and the causes of the phenomenon, due to the traditional severance of glamping from tourist behavior. Therefore, this study focuses on this emerging tourism phenomenon and attempts to illustrate the scenic dimension of glamping through the aesthetics of atmosphere

theory. The perceived value of travelers' camping experience is examined within the SOR framework to gain a fresh and nuanced understanding of glamping.

### 2.2. The Aesthetics of Atmosphere

In this new era, people's desire for a better life has intensified and Welsch points out that people's perceptions of aesthetics are changing and that the aesthetics of objects are as important as their practicality [13]. Based on the increased emphasis on aesthetics, the German scholar Gernot Böhme developed the aesthetics of atmosphere theory. He defines atmosphere as "the definite presence of something" and notes that the term "atmosphere" is increasingly used as a metaphor for a particular mood that pervades space. The aesthetics of atmosphere focuses on the human experience in the environment, taking into account the relationship between the human condition and the nature of the environment [14]. The aesthetics of atmosphere was first established as a sub-discipline of philosophy by Alexander Gottlieb Baumgarten and later integrated into the new system of phenomenological theory through Hermann Schmitz's concept of atmosphere. On this basis, Böhme combines the concept of atmosphere with the general theory of perception, using the receptive body as a channel and embedding perceptual patterns and sensory functions into the environment. All "atmosphere-making" activities are considered aesthetic activities, existing between subject and object, perceiver and perceived. Simply put, "atmosphere" is created by the interaction of objective factors (such as lighting, music, and props) and subjective factors (the people on stage, the behavior of the performers, and the audience's sense of immersion). The atmosphere as an intermediate quality cannot be separated from its surroundings [15]. In the current context, the atmosphere can be defined as a mood space with specific emotions [16], which explains how the camping environment affects us through the body to produce the beauty of the atmosphere and the sense of romance.

Böhme believes that people can create emotions, alike the stage, which is at its origin a "commodity aesthetic", and that, through staging, the atmosphere can be created as a practical expression of the aesthetics of atmosphere in our lives. As festivals have a festive atmosphere and roses have a romantic one, this idea emphasizes the subjective perception of aesthetics (i.e., how one feels in the presence of things), with body, presence, and space being three important dimensions surrounding the aesthetics of atmosphere [17]. Previous research has shown that there are two ways of perceiving atmosphere: ingression and discrepancy. The feeling of physical and mental pleasure that one enjoys when in a breezy spring environment is the ingression of the atmosphere, while the inability of a sad person to be infected by the atmosphere and experience the happiness of spring is a discrepancy [18]. So, how is the atmosphere of nothingness perceived? Lefebvre used three concepts to mark out a phenomenological approach to the production of space in three dimensions—the perceived (perçu), the conceived (conçu), and the lived (vécu) [19]. In this, the space of perception can be apprehended by the senses. Perception forms an integral component of every social practice. It encompasses everything that is presented to the senses. Not only sight but also hearing, smell, touch, and taste. The perceptible dimension of space is directly related to the materiality of the 'elements' that constitute 'space'. Therefore, the atmosphere can be perceived and created [20]. Tourist places can create a unique perception and atmosphere through light, color, taste, fog, sound, cultural icons, and symbolic objects, as in glamping experiences, wherein tents, canopies, barbecues, strings of lights, scenic beauty, lawns, and skies create a strong and pleasant atmosphere. Therefore, campsites can use unique cultural elements to elevate the scene to a space with depth of meaning and emotional tendencies created by the aesthetics of atmosphere, enhancing the tourist's perception of the experience through a holistic physical experience [21].

For glamping, the atmosphere acts as a general scenario wherein the campsite is transformed from an external appendage of things into an internal dimension that structures them. Things not only inhabit the space they physically occupy, but also the space that extends out from and is impregnated by them. On the one hand, the campsite creates an atmosphere through symbols, fields, and metaphors; on the other hand, the tourist

experiences the atmosphere through embodiment and empathy. The existing research has rarely clarified the dimension of atmospheric aesthetics at a theoretical level and relatively few studies have focused on the role of the atmosphere in the construction of scenes for tourism. Based on this idea, the specific objectives of this study include (1) examining the typical imagery of glamping and capturing the unique imagery of glamping using user-generated content (UGC) data; (2) dissecting the scenic dimension of glamping using grounded theory, relying on the aesthetics of atmosphere theory; and (3) using the SOR framework as a tool to clarify the response mechanisms between the elemental levels of glamping.

## 3. Methodology

### 3.1. Research Method

This study adopts a qualitative research method to analyze the factors that together create an atmosphere scenario of glamping and the propagation behavioral chain. Qualitative research is applied to explain how this occurs [22]. Firstly, a grounded theory approach is used to identify the conceptual dimensions of glamping and to analyze the process of forming a sense of atmosphere. Secondly, using content analysis, word frequency statistics for specific words are formed by the coding of qualitative text and photo content [23]. To sort out the unique image of glamping. Finally, inductive analysis is used to explore the path from online communication to the offline quality development of glamping; to sort out the inner relationship between the physical environment, situation, and context; and to establish a communication model for the quality development of glamping according to the experiential marketing path. To analyze the law of glamping atmosphere scene propagation.

### 3.2. Data Collection

The data sources for this study include both online data from the Red app and offline interview data. The internet is an important factor in the development of the glamping market as well as an important marketing management tool in enabling glamping practices to achieve and maintain a competitive advantage. Considering that the development of glamping relies to a certain extent on new media platforms, it is worth pointing out that since the Red app is the main place where information on glamping is shared in China, and its sharing notes are detailed and abundant, this study chose the Red app as the platform through which to collect the research sample. According to the principles of theoretical sampling, the selected sample should meet the requirements of theoretical construction [24], meaning it must be able to answer the given research question. In terms of the content of the notes, the following selection criteria were finalized for the study sample, taking into account the role of conscious guidance versus unconscious sharing and the variability of different campsites:

(1) Sharing of glamping content, which may include either photo or text content or both.
(2) An excellent ability to articulate the construction of glamping scenes and the psychological activity of the tourist.
(3) A detailed description of the perception of the atmosphere obtained after a glamping experience.

Based on this criterion, we have selected 80 popular notes on the Red app that were publicly posted and involved the sharing of glamping content. This includes a representative selection of glamping promotional content published by the official public website as well as the top 20 individual shares in terms of favorites, likes, and comments. The period for the publication of posts considered is from June 2020 to June 2022, with a total of 87,884 words. In addition, the large number of images contained in quality notes serves as important data, as images are an important part of UGC and are ideal for analyzing the construction of a glamping atmosphere scenario. A total of 737 images were collected for the study. To enrich the categories and concepts constructed [25], interviews were conducted with practitioners of glamping, including campground owners, campground staff, and

well-known camping bloggers, see Table 1 for details. In-depth interviews were conducted in July 2022, after contacting interviewees by private message through Little Red Book, learning of their willingness to be interviewed, and receiving their consent. The interviews were based on questions such as "What do you think about the explosion of glamping?", "What drives the public to participate in glamping?", "What activities do campsites offer for campers?", "What do people like to do while glamping?", and "What do campsites and content about glamping communicate to people?". The interviews were limited to roughly 30 min and were translated into text totaling 9365 words. The qualitative analysis software NVivo 11 and Rost CM6 were used to explore the scenic elements of the glamping experience and the perceived attributes and content of the traveler. The diversity of data sources ensured complementarity across the data set, enabling the continuation of the data collection process until the concept became sufficiently saturated to permit theoretical development, thereby allowing for a clearer idea of the antecedents and consequences in the sustainable development of glamping.

**Table 1.** Basic information of respondents.

| Interviewee | Gender | Age | Occupation | Proportion |
|---|---|---|---|---|
| 1 | Male | 26 | Staff | |
| 2 | Female | 34 | Staff | |
| 3 | Female | 20 | Staff | 50% |
| 4 | Male | 25 | Staff | |
| 5 | Male | 56 | Campground owners | 12.5% |
| 6 | Male | 33 | Blogger | |
| 7 | Male | 28 | Blogger | 37.5% |
| 8 | Female | 21 | Blogger | |

Of the respondents, 50% were campsite staff, 12.5% were campground owners, and 37.5% were famous camping bloggers in the Red app. Among them, 62.5% were male and 37.5% were female. In terms of age, young adults in their 20s and 30s dominate, accounting for 87.5%, which is related to the fact that glamping is a sunrise industry in China.

*3.3. Grounded Theory Analysis*

Grounded theory is a qualitative research method that generates concepts through the analysis, collation, and induction of primary sources, continuously compares sources and concepts, refines and generates theoretical questions, establishes links between categories and main categories, and forms theories from the bottom-up [26]. The reasons for adopting the grounded theory are as follows. Firstly, the research on glamping scenes is still at the beginning stage, with no mature constructs, unclear categories and dimensions, immature measurement scales, and a weak theoretical foundation. Therefore, it is not convenient to use quantitative research methods directly. Secondly, this study focuses on the process of tourists' perceptions of the ambient scene of the campsite. The questionnaire survey was difficult to fully tap into the real thoughts of the tourists. However, qualitative research methods, as represented by grounded theory, apply to answering the "what" and "how" questions [27]. Finally, in contrast to other qualitative research parties, grounded theory can exemplify the process of theory formation from the bottom-up [27]. At present, the construction of the glamping ambiance is still a new concept and lacks a systematic research reference. Before the study, it was not possible to precisely define the specific dimensions and actual performance of the glamping ambiance scene, nor was it possible to understand the process, characteristics, and internal logic of the construction of the glamping ambiance scene. Given that grounded theory emphasizes theoretical constructs under material analysis, it is suitable for exploring and analyzing relatively new fields [28]. Therefore, this study chose to follow the procedures and methods of grounded theory to analyze, organize, summarize, and code the raw data [29], which was to achieve the process of theorizing the glamping atmosphere scenario.

In coding with the use of grounded theory, we focused on the naturally occurring "units of meaning" in the text and marked them while placing the construction of textual

material and conceptual dimensions in open and continuous comparison, merging, revising, and recategorizing new codes as appropriate until the code was saturated [30]. A total of 80 notes were used for this process and interviews were later added for saturation testing until no new codes emerged. The coding process was based on three coding levels—open coding, axial coding, and selective coding [31] (Table 2)—as follows.

**Table 2.** Coding process for the sense of glamping atmosphere.

| Open Coding | Axial Coding | Selective Coding | Code Meaning | Example Text |
|---|---|---|---|---|
| Snowy mountains, lakes, beaches, forests, seashores, lawns, cliffs, waterfalls, deserts, valleys, wetlands, streams and springs, countryside, rice paddies, parks, rooftops | Site selection | Physical environment | Glamping scenes construct objective scenery or situations that are represented as concrete visual images. | *The exclusive snow-capped campsite offers the luxury of seclusion in front of a campfire, in a tent, under a canopy, or beside a lake, while at night, the crackling of the wood fire echoes the stars in the night sky, providing a quiet and relaxed atmosphere.* (XHS-2) |
| Bouquets, string lights, bonfires, ethnic decorations, woven baskets, dolls, bubble machines, swings, magazines, stereos, camping lights, balloons | Embellishment | | | *Add to that a set of titanium Snow Peak mugs, a Bialetti mocha pot gurgling and smoking, and a B&O Bluetooth speaker pressed against a Popeye magazine playing city pop.* (XHS-8) |
| Coffee pots, cassette stoves, tents, picnic mats, canopies, egg roll tables, moon chairs, barbecues, toilets, toiletries, toiletry sets, campfires, ingredients, cutlery, inflatable mattresses, tidy boxes, pull carts, hammocks, sleeping bags, outdoor power supplies, mosquito repellent | Glamping equipment | | | *The camping equipment was rented for a lazy glamping experience. We went empty-handed and booked with a boss in the Western Hills who had all the equipment for her campsite (canopy, camping tent, tables and chairs, ovens, picnic cloths) at a very affordable price.* (XHS-7) |
| Chatting, night running, wood chopping, fireworks, drones, skateboarding, paddle boarding, frisbee, open-air cinema, barbecue, bands, tea, photo-taking, books, wild fishing, drinking, kayaking, motor boating, karting, kite flying, board games, rock climbing, archery, DIY, cycling, hot air ballooning, swimming, stand-up comedy, KTV, paragliding, bazaar | Activity scene | Situation | The specific contextual expression of glamping is the combination of activity and nature, characterized by the integration of emotion into the landscape. | *Enjoy the birdsong, gurgling water, green grass, tents, beer, barbecues, bands, and open-air films.* (XHS-10) |
| Nature, night, summer, autumn, spring, outdoors, waves, insects, evening sun, sunshine, sunrise, rain, breeze, moonlight, blue sky, stars, sunset | Event environment | | | *Sleeping at night to the sound of the waves and waking up in the morning to the sound of the waves is especially good for the whole family and a different kind of outdoor experience.* (XHS-16) |
| Open, quiet, comfortable, vintage, warm, ambient, carnival, surprise, novelty, warm, cozy, romantic, nice, love, affordable, convenient, clean, hygienic | Visual impression | Context | The presence and movement of the tourist's mind in time and space after experiencing glamping is expressed in aesthetic moods and life lessons. | *It was so nice to go glamping with the people we love. The gift of nature was so new to us, full of greenery, mountains stretching to the sky, and a quietness I hadn't seen for so long that I didn't want to leave.* (XHS-13) |
| First time, family, parent–child, couple, friend, pet, party, holiday, punchline, birthday, reunion | Memorial sense | | | *Wanting to take my girlfriend out and relax after the epidemic, I chose to enjoy an urban glamping trip by the Golden Rooster Lake. As the first outdoor activity after the epidemic, it was a treasure trove of picnics, camping, kite flying, and pets, adding fullness and joy to our life memories.* (XHS-14) |
| Purity, freedom, recovery, escape, back to nature, forgetting troubles, emotional resonance, back to basics, the unforgettable, carefree, passage of time, precipitation, environmental protection | Depth perception | | | *It was also my first experience of the purity and pleasure that comes with wild glamping. To be so close to nature, to return to its natural essence without losing the pleasure of enjoying an afternoon tea with a cup of delicious coffee that resonates with the originality and the soul.* (XHS-5) |

Open coding: Through repeated reading and analysis, the content of the collected notes and the interviews were labeled according to the principle of coding extracted close to the material. In the process of annotating, naming phenomena, and continually comparing the contents of the texts, the units of meaning found in the texts were treated as separate symbolic codes. After two encodings and several corrections, the final 139 original open codes were extracted.

Axial coding: The main task of the axial coding phase is to explore and establish the organic links between the various conceptual categories implied in the basic codes identified in the open-coding phase. The search for similar phenomena led to the continuous discovery of categories and the naming of similar phenomena with a unifying concept, culminating in eight major categories based on the categorization of open codes: site selection, embellishment, glamping equipment, activity scene, event environment, visual impression, memorial sense, and depth perception. The choice of the campsite is often closely linked to nature and, whether the visitor chooses the mountains or the sea, the overall ambiance of the environment in which the campsite is situated provides the basis for a sense of glamping. The embellishments and facilities of the campsite are strongly symbolic and are an intuitive experience for the traveler as a result of the sensory stimulus response. The open-air movie, frisbee, and fishing activities are mapped by the waves, insects, and sunsets and they participate in building a distinct context for glamping activities, achieving a blend of contexts centered on satisfying the senses of tourists. Glamping offers travelers the opportunity to spend time with family, friends, and pets, with wild nature and carefree time being the recurring focus of glamping enthusiasts.

Selective coding: In selective coding, site selection, embellishment, and glamping equipment are relegated to the level of the physical environment, creating the tourist's first opportunity for decoding and the perception of the objective environmental experience of glamping. The activity scenes and event environment constitute a situation that links geospatial, landscape, and glamping experiences, which is a continuation of the presentation of the concept of the physical environment. Context is the subjective spiritual feeling projected in the physical environment and the situation within the tourist, the fit between the previous perception of glamping and the tourist's expectations. Travelers have individual differences in their understanding of the glamping atmosphere and the sublimation of the mood is a spiritual pleasure for travelers that allows them to transcend their senses.

## 4. Data Analysis

### 4.1. High-Frequency Word Analysis for Glamping

After identifying the dimensions of the construction of an aesthetic atmosphere for glamping, the path from online communication to the sustainable offline development of glamping became the next focus of research. This research was explored through content analysis of the interview and note texts, starting with a word frequency count of key glamping constructs in the texts (Table 3) to portray typical atmospheric scenes for glamping.

Table 3 shows that the terms "nature", "experience", "outdoors", "sophisticated", and "equipment" appear frequently and are visual representations of glamping perceptions. In general, campsites are intentionally located in areas of natural beauty, with "mountains and water" and "waves and sand" providing the first layer of "sophistication" to glamping, in line with the previous research by Filipe et al. [32]. In addition to rural and suburban campsites, urban campsites also attract travelers with their "convenience" and "affordability", with "green spaces" in cities and "rooftops" in high-rise buildings allowing glamping enthusiasts to take a short break. Glamping equipment plays a decisive role in creating a sophisticated atmosphere, with wild luxury elements such as "canopies", "tents", "cassette stoves", and "coffee machines" prominent in the notes and photos shared. The close contact with nature and the special outdoor experience is therefore the underlying emphasis of glamping. Second, the "glamping +" approach reaches different interest groups, relying on

the campsite to host various activities, providing a venue for "music festival" fans, "movie" fans, and "outdoor leisure activities" fans and has become an excellent platform for leisure and social activities. The natural environment, activities, facilities, and equipment at the campsite provide a sense of comfort and well-being, echoing the study by Lu et al. [6].

**Table 3.** High-frequency vocabulary of glamping.

| No. | Keywords | Frequency | No. | Keywords | Frequency |
|-----|----------|-----------|-----|----------|-----------|
| 1 | Glamping | 696 | 16 | Atmosphere | 81 |
| 2 | Nature | 279 | 17 | Enjoy | 78 |
| 3 | Experience | 219 | 18 | Romantic | 72 |
| 4 | Outdoor | 195 | 19 | Canopy | 72 |
| 5 | Exquisite | 163 | 20 | Sunset | 66 |
| 6 | Equipment | 144 | 21 | Chengdu | 57 |
| 7 | Tent | 117 | 22 | Convenient | 52 |
| 8 | Friend | 102 | 23 | Music | 51 |
| 9 | Greenbelt | 96 | 24 | Parent–child | 39 |
| 10 | Take pictures | 94 | 25 | Forests | 34 |
| 11 | Barbecue | 93 | 26 | Outdoor movie | 34 |
| 12 | Weekend | 89 | 27 | Lawns | 34 |
| 13 | Seaside | 87 | 28 | Wonderful | 33 |
| 14 | Summer | 82 | 29 | Starry sky | 32 |
| 15 | Coffee | 82 | 30 | Healing | 30 |

The photographs provide a more direct representation of the physical and emotional context of a glamping trip, dissecting the elements contained in the photographs and coding each photograph as a separate unit of content analysis. The purpose of coding is to parse the sample data into different units of meaning and transform them into multiple labels, which can then be reclassified to facilitate the comparison of similar things [33]. Specifically, a project was first constructed in NVivo 11, into which sample photographs were imported, the content features of the photographs were encoded so that the attributes gradually emerged from the material [34] and each encoded attribute was named (e.g., "lawn", "canopy", "people") and marked as a free node. Due to the complexity of the photographs themselves, the content of the photographs in this study was "shredded" into multiple elements and encoded into multiple nodes, totaling no more than four [34]. Special imagery related to camping was incorporated into the coding system: for example, the photo in Figure 1 shows depictions of a "picnic mat", "outdoor trolley", and "egg roll table." After coding all the photos, the concepts and contents associated with the nodes were further examined; next, the free nodes were classified and merged to refine the themes found within them and then the pre-generated free nodes were coded level by level under the child and parent nodes to form a tree node [34]. For example, "tent" and "starlight" are merged into the parent node "physical environment" and each child node can only be merged into one type of parent node [34]. The main images of the glamping imagery from the tourist's perspective are summarized in Table 4.

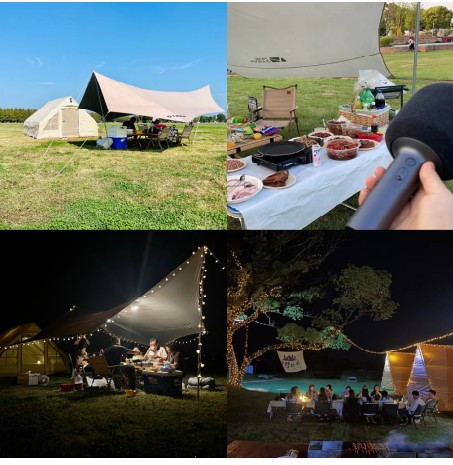

**Figure 1.** Example of a glamping picture.

**Table 4.** High-frequency noun statistics for glamping image metadata.

| Parent Node | Percentage | Child Nodes | Key Elements | Frequency |
|---|---|---|---|---|
| Physical environment | 54.31% | Site selection | Lawns | 309 |
| | | | Lake | 206 |
| | | Embellishment | Starlight | 128 |
| | | | Canopy | 302 |
| | | | Coffee machines | 289 |
| | | Glamping equipment | Tents | 265 |
| | | | Cassette ovens | 223 |
| | | | Moon chair | 219 |
| | | | Trolley | 107 |
| Situation | 25.56% | Activity scene | Screen | 202 |
| | | | Barbecue | 162 |
| | | | Board games | 73 |
| | | | Bonfire | 70 |
| | | | Fishing | 68 |
| | | Event environment | Blue sky and white clouds | 331 |
| | | | Night view | 58 |
| Others | 20.13% | | People | 383 |
| | | | Food | 242 |
| | | | Pets | 81 |
| | | | Map markers | 53 |

The high-frequency attributes in the photographs can be used to guide the design and management of glamping experiences [35]. As Figure 1 and Table 3 show, the analysis of high-frequency nouns in the metadata of glamping images shows that the physical context is the main source of the visual impact. The use of the campsite's natural environmental conditions along with sophisticated equipment and decoration to carefully set the scene and export the aesthetics of living in a natural environment forms the basis for a sophisticated glamping atmosphere. Further, the sense of atmosphere is reflected in the context, wherein indoor activities are mostly restricted and people move to the outdoors, where the experience echoes the atmospheric background. Barbecues, open-air movies, and water play are set against an atmospheric backdrop of sunshine, evening sun, and even drizzle, presenting a very different and more healing experience than that of the city scene, according to the respondents. In addition, there are numerous close-ups and group shots of people, food, and pets, including both interactive snaps and focused closeups, conveying the camper's love of life and yearning for nature. Additionally, there is a large number of campsite maps among the images in the sharing notes, mainly of newly discovered, niche campsites, which provide effective evidence of glamping going from online to offline.

*4.2. Analysis of the Dissemination Process of Glamping*

The stimulus-organism-response (SOR) model is a theoretical model proposed by environmentalists Mehrabian and Russell to investigate the effects of external environmental stimuli on an individual's cognitive or emotional responses and to further predict responsive behavior [36]. The term "stimulus" refers to the external environmental factors, "organism" refers to the cognitive and emotional responses of individuals stimulated by the environmental factors, and "response" refers to the behavioral responses of individuals based on the cognitive and emotional changes that occur [37]. The framework suggests that external environmental factors (stimulus) affect people's internal states (organism), which in turn leads to convergent and avoidant behavior (response). Specifically, external environmental stimuli trigger an individual's cognitive and emotional state, which in turn triggers the individual to adopt a corresponding behavioral response [38]. Because the SOR framework integrates cognition, emotion, and behavior and better reflects the process of generating individual behavior, it is widely used in several disciplines such as environmental psychology, organizational behavior, marketing, and tourism [39]. In particular, in recent years, tourism scholars have used the SOR framework to explore the impact of destinations on tourists' emotions and behavior [40,41]. In fact, in a tourism context, external stimuli include both objective factors such as the landscape of the destination and subjective factors such as the availability of various services, activities, and the reputation

of the destination. They all influence the emotional experience of the traveler, which in turn stimulates a corresponding behavioral response [42].

Glamping brings a rooted connection that, as a scene, creates excitement; as an activity, it triggers cognition and emotion. Through the stimulation of the physical environment it provides, glamping offers a way to deepen the senses of sight, hearing, taste, touch, and smell; to expand one's cognition and emotion with the context and to match the stimulus with the body's response to influence its behavior. That is, travelers spontaneously share and advise one another on glamping experiences through texts and pictures that are continuously dispersed on social media. In this process, the tourist shifts between the roles of online observer, offline experiencer, and online sharer to complete a cycle of tourist gaze [43].

*4.3. Analysis of Stimulus Elements*

The analysis of the data shows that the physical environment and situational elements explored in this study are the stimuli that conjure the experience of glamping. For example, the participants shared the following descriptions: "*The glamping ambient light, the main light, and the string light, all bring a better experience to glamping*", said one participant (XHS-3). "*The campsite has a beautiful view, the tent is clean and tidy; in addition, there are small animals such as bunnies, Alaska, corgis, cows, and pet pigs in the campsite; the ambiance is full*", said another (XHS-4). "*The beige and white color scheme makes it look warm and inviting. It is decorated with greenery, rattan and silk textures, pottery and wood carvings, and is simple and rustic*", offered another respondent (XHS-21). Based on the visitor-generated content, a typical image of glamping can be drawn: a long table under the canopy filled with cups, plates, dishes, and drinks, next to the set-up tent and the hot meat skewers sizzling on the barbecue. Children playing on the lawn, adults sipping coffee, and chatting on moon loungers—this is the kind of atmosphere that attracts more and more people to glamping. Furthermore, glamping offers an "immersive" leisure path in which people take the opportunity to experience a variety of leisure activities in nature. "*We recommend bringing a change of clothes and flip-flops, and children can play in the sand and water. The current at the shallow end is not too strong either, so it's good for taking small children*", stated one participant (XHS-7). "*This glamping site fulfills everything you can imagine for your dream holiday. You can try fishing, pot-throwing, frisbee, archery, BBQ, fruit-picking, and more over here, and on weekends, gather for open-air movies, cold fireworks, bonfire parties, and more*", said another (XHS-22). "*Who says rain isn't good for glamping, with white noise in 4D surrounds and the good-smelling air that fills the rain-washed mountains? A favorite song plays on the stereo and the sound of rain pattering outside the tent is another accompaniment*", said yet another (XHS-20). People sometimes set up camp in groups and build new communities for a limited time. The changeable weather and the abundance of activities condensed into a "situation" perceived by the traveler provides a degree of drive to glamping.

*4.4. Analysis of Organism Elements*

The "organism" of the traveler is transformed by the unique physical and situational stimuli of glamping experiences, with the context serving as the primary organism element in the analysis that reflects the mental perception of the traveler after the experience of glamping. Specifically, positive emotions with monumental meaning, visual impressions, and perceived aesthetics are what provide spiritual support for the subsequent development of glamping. "*We are on the opposite side of town, relaxed and romantic*", said one participant (XHS-10). "*The nights of glamping in Cangli are accompanied by warm yellow lights, very atmospheric, with a pure and noble feeling that purifies the soul. Each tent set up is inscribed with a different memory, with infinite beauty and tranquility for oneself!*" said another (XHS-13). "*In fact, exquisite is never glamping; it should be living, bringing life to the outdoors, going outdoors, spreading the wild, enjoying the sense of freedom away from the hustle and bustle!*" exclaimed another (XHS-19). Through the dissemination of social media, the "sense of atmosphere", "ritual", "romance", and "freedom" are the summation of the tourist's perception that

emerges through the phenomenon of glamping—the physiological and chemical pleasure in the smell of mud, the recognition of a more vivid self, a utopia where the spiritual world of modern people shines into reality. The stimulus—organism elements observed in this study through user-generated content and in-depth interviews—are similar to the findings of Ana and Fernando's exploratory study of glamping [7]. The expressive dimensions of visitor experience focus on the supply elements of the campsite and the elements perceived by the tourist; these feature terms are consistent with the positive expressive dimensions in previous studies [2,44,45].

*4.5. Dissemination Path Analysis*

Through text analysis, keyword extraction, and analysis of theoretical elements, the SOR framework has gradually become clear in the process of glamping communication. In the actual development process, either by the direct stimulation of objects, situations, or contexts to generate sharing and communication reactions or through a complete chain of reactions that prompt the sharing and spreading of information, the proliferation of glamping gains obvious advantages from social media platforms.

Among other things, the environment, facilities, and equipment of glamping facilities help travelers to share their experiences. "*Some tips: have a rain jacket and a pair of rain boots to keep you dry and keep you looking good. Camping lights are essential for night lighting!*" said one (XHS-31). "*Navigate by car in the last picture; you can park on the side of the road and a little further in is the lakeside lawn! As a bonus, the toilets are very handy, right next to each other, and clean!*" said another participant (XHS-16). Being close to nature and enjoying enhanced social interaction are the inherent attributes of glamping and, at the very beginning of the construction of their perceptions of the physical realm of glamping, travelers can keenly explore its symbolic characteristics and spread them through the internet. Gradually, glamping has become a strong demand of modern urbanites as a backdrop for tourism activities, outputting a rich mix of activities in both the limited space of the campsite and the panoramic outdoor setting. "*Gaining wood-chopping skills, sipping a cup of autumnal hand-ground coffee, and experiencing your own 'yearning life'*" are several such activities, said one participant (XHS-4). "*Taking part in the Star Festival under the YuYang Mountain was an extraordinary experience*", said another (XHS-17). Yet, another described "*Running around on the lawn, throwing frisbees and starting a bonfire at night with the stars for a summer garden party*" (XHS-23). People immerse themselves in the pleasures and relaxation of glamping by engaging their five senses and many new or niche activities are springing up into the glamping scene, becoming fodder for shared travel experiences. More often than not, travelers condense the experience into a sense of mood and their positive emotions contribute to positive word-of-mouth communication. Glamping arises from the desire for an overnight utopian experience, an ideal state of living, and an expression of freedom. "*Glamping alone is a romantic enjoyment; slowing down is more healing and warming*", one participant explained (XHS-9). Another said, "*The first time I tried to go glamping with my children, there was no panic as expected; the children helped us to set up the tent and string vegetables together. The long-lost nature contact made the children feel very happy*" (XHS-12). "*Looking up at the romantic starry sky, enjoying the cozy life at will, the camping tent is supported by their poems and faraway places*", described another participant (XHS-30). With this view, this study proposes a conceptual model based on the aesthetic theory of atmosphere and the SOR model, as shown in Figure 2.

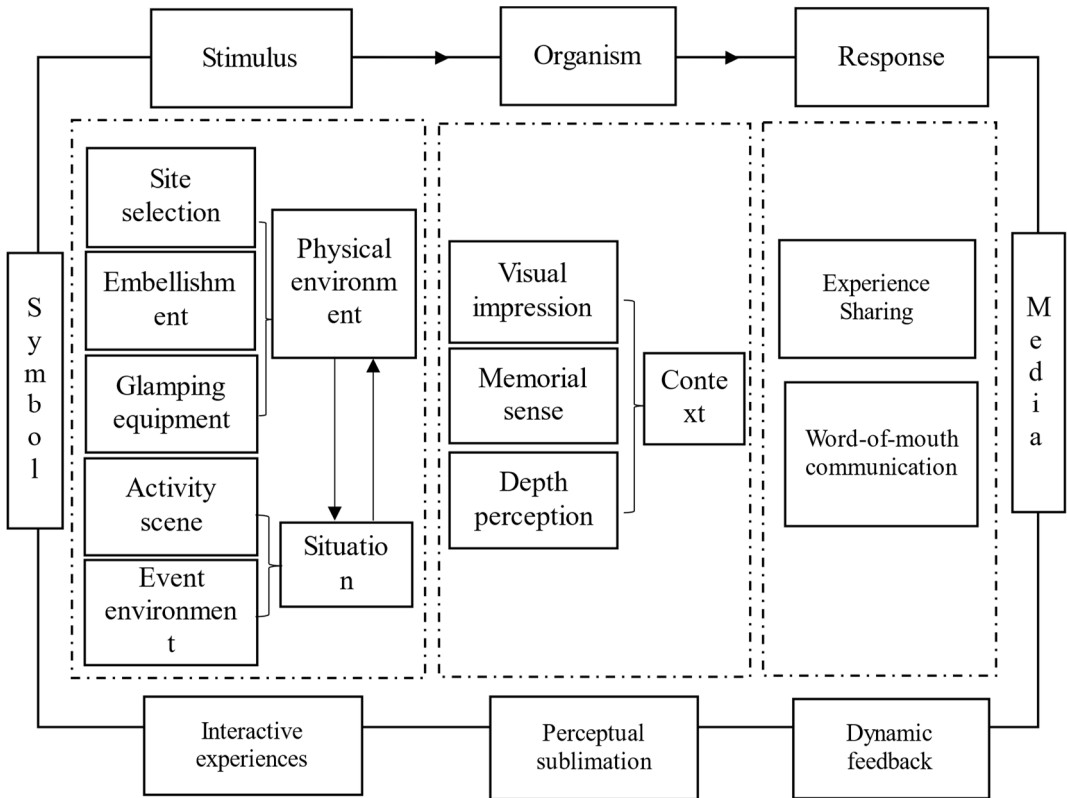

**Figure 2.** Glamping's online–offline sustainable development communication path.

## 5. Discussion and Conclusions

This paper constructs a communication pathway for the high-quality development of glamping based on online graphic material and semi-structured interviews. Atmospheric aesthetics is used to construct the dimension of a glamping scene and the communication path is analyzed by the stimulus-organism-response theory. The main findings of the study are as follows.

First, this study examines the emerging tourism activity of glamping, through an analysis that provides an in-depth understanding of three important dimensions of the glamping scene, exploring the influence of atmospheric perception on the dynamic feedback from tourists in terms of the physical environment, situation, and context. The study found that the element of stimulation is still an important support for the development and dissemination of glamping and that the physical environment, based on site selection, embellishment, and glamping equipment, and the situation, which is a combination of activity scene and event environment, serve as the basis for the condensation of symbols. As a spatial representation of the atmosphere, the rendering of the interactive experience creates a positive response from the tourist. Under the effect of organism perception, visitors find joy, happiness, and meaning in glamping. This in turn promotes a feedback response that sets off a crescendo of experience sharing and word-of-mouth communication.

By analyzing the physical environment, we found that glamping decoration, equipment, and activities were the main focus of visitors' attention. In China, while enjoying nature, visitors need the sense of atmosphere and convenience that comes with modern facilities and equipment. More visitors simply want a backdrop for photos, so glamping sites offering elements such as starlights, cassette stoves, and campfires have emerged, contributing to glamping becoming a popular tourist activity for the masses. On the other hand, glamping is more demanding of the environment, in a more casual way, and its symbolic meaning is more prominent. The structure is found in the glamping scene that upgrades the traditional camping experience for consumers with strong social and comfort needs. Glamping enhances the interactive experience of campers by combining

niche outdoor activities such as frisbee, paddleboarding, and other outdoor sports; music festivals; campfire parties; and other comprehensive activities to meet the consumer's need to highlight their social status through their refined taste in culture in a situational atmosphere, directly prompting them to consciously spread this aesthetically pleasing lifestyle. In short, glamping combines the aesthetics and personal lifestyle sought by travelers with a higher level of comfort, more style, and the 'ritual' that travelers seek in terms of decorative equipment and activities that are unique to glamping. The campsite as a representational space embodies a complex symbolic role, a space for campers and camp owners, a space for passive experience, and a space for imagining attempts at transformative debugging. Therefore, the supply side needs to start with the physical environment and situation, strengthen their creation for optimization of glamping, and upgrade the camping experience from the whole system of site selection to facilities, equipment, food, and activity content. Amplifying travelers' closeness to nature and love of life provides the basic impetus for the continued development of glamping [7].

Second, the "organism" perception is the key element in developing and disseminating the "online–offline" progression of glamping; the context is the emotional sublimation of the traveler's bonding with glamping. Behind the explosion of glamping is a new generation of travelers dissecting and resetting themselves, as the outdoors allows one to leave the familiar [46] and, through the "isolation" of the scene, the individual is allowed to perceive the real. Whether they are focused on following trends or fashions, meeting social needs, satisfying their vanity, or just relaxing and enjoying life, all will find a projection of their desires in glamping. The physical and situational interaction of the campsite is powerful. The novelty of experiencing something for the first time, the time spent interacting with family and friends, and the immersion of being in the scene all bring mental satisfaction to the camper. In addition, in China, the rapid development of glamping in the wake of COVID-19 has led to a different understanding of life and living, with a rich sense of experience acting as a significant driver for the conscious spread of glamping.

Finally, the dynamic feedback of "online–offline" is achieved through experience-sharing and word-of-mouth communication. With the condensation of the elements of stimulus and organism, campers can spread the scenery, facilities, activities, and feelings of glamping on social media platforms, enriching the scene with unique atmospheric and romantic touches and deepening their perception of the experience at the moment or after glamping. This corresponds to the open and reorganized interaction between the concrete and the abstract, the real and the imaginary in the "space of perception", the "space of conception", and the "space of life" [47]. Exquisite campsites can therefore dig deeper and integrate packages that include accommodation, facilities, and activities to meet the camper's desire for an atmospheric scene that emphasizes the camper's enjoyment of nature while providing an opportunity for social entertainment. The satisfaction of special needs being met, such as pet companionship, niche sports, and festive extravaganzas, is conducive to sharing and spreading the word after the camper has enjoyed the novelty of the experience. In particular, the Red app's shared notes and the responses from interviewees indicated that the strong desire to participate in glamping began with following others' notes on social media and that the complete cycle of "watch online, experience offline, share online" was typical in the sharing of glamping experiences. In a world where everyone is a marketer, campsites and media platforms encourage campers to share descriptions of their experiences to help spread the word about glamping, this helps glamping tourism to be developed deeply and sustainably.

In summary, this study first answers the question of what the atmospheric scene of glamping is. The phenomenon of glamping being very atmospheric is linked to the aesthetics of atmosphere theory. Using grounded theory and content analysis to clarify the atmospheric scene dimension of glamping and to sort out the typical image of glamping. Secondly, glamping has become a hot content on social media. Based on the glamping scenario dimension of question one, the communication path of glamping was constructed through the SOR framework. The organic combination of the aesthetics of atmosphere and

the SOR framework broadens the theoretical boundaries of the aesthetics of atmosphere. In practical terms, analyzing the key elements shared and communicated by visitors can guide the development of glamping and be an effective reference for the construction of campsites and green parks around the city.

## 6. Research Gaps and Prospects

First, in this study, when the web data were collected, due to the influence of the software's precise recommendations, the content of the notes was found to center on our geographical location after sorting by both hotness and time, so there were more notes in and around Suzhou, resulting in the findings of this paper having some regional limitations. Second, this study uses positive tourist emotions as the main reference point and does not consider the negative perceived effects of glamping by tourists, resulting in the findings of this paper having some content limitations. Third, this paper uses qualitative analysis to conduct research, lacking the test of quantitative analysis. In the future, quantitative research can be conducted based on the design of a scale founded on the rooting results to further enrich the research results on the development of glamping.

**Author Contributions:** Conceptualization, T.S.; methodology, T.S.; software, T.S.; validation, T.H.; investigation, T.S.; data curation, T.S.; writing—original draft preparation, T.S.; writing—review and editing, T.H.; project administration, T.H.; funding acquisition, T.H. All authors have read and agreed to the published version of the manuscript.

**Funding:** This research was funded by the National Social Science Foundation of China, grant number: 17BGL123.

**Institutional Review Board Statement:** The study was conducted in accordance with the Declaration of Helsinki and approved by the Institutional Review Board.

**Informed Consent Statement:** Informed consent was obtained from all the subjects involved in the study.

**Data Availability Statement:** Anonymous verbatim of the interviews are available on request.

**Acknowledgments:** Thanks to Gao Lihua of Soochow University for her advice.

**Conflicts of Interest:** The authors declare no conflict of interest.

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
