# Peer review of "Research of Glamping Tourism Based on the Aesthetics of Atmosphere"

_sustainability, doi:10.3390/su15010581_

Round 1

Reviewer 1 Report

Dear Author(s),

Thank you for sharing your research findings with us.

The manuscript deals with an interesting topic - Sustainable Development Research of Glamping Tourism Based on the Aesthetics of Atmosphere. Both the methodological part and the results are solid. I would have few suggestions for the authors. I think the title is too long, perhaps the phrase "sustainable development research" could be shortened. The authors indicated that they conducted interviews with glamping practitioners, including campground owners, campground staff, and well-known camping bloggers. However, there is no information on how many interviews were conducted, and it would be valuable to know the structure of the interviewees, i.e., what is proportion of practitioners, campground owners, staff, and others.

The results obtained by the authors are quite as expected and the specificity of glamping compared to traditional camping is not apparent. It would be good to see this explained in more detail.

I found the language of writing pleasant and easy to follow.

Good luck with your research.

Reviewer 2 Report

This article examines the typical imagery of glamping and captures the unique imagery of glamping using UGC data; dissects the scenic dimension of glamping using grounded theory, relying on the aesthetics of atmosphere theory; and (3) uses the SOR framework as a tool to clarify the  response mechanisms between the elemental levels of glamping.

In the literature review part, the author(s) state that “Previous research has shown that there are two ways of perceiving atmosphere: ingression and discrepancy”. They have a weakness in reviewing the related theories about that, such as the lack of discussing Henri Lefebvre’s perceived, conceived, lived space conceptualizations. And more.

The main subject of the research is so interesting and they design a quantitative research technique to test the impact of digital transformation on the international strategy of Chinese Enterprises by applying a multiple regression model. NVivo 11 and 205 Rost CM6 was used to explore the scenic elements of the glamping experience. In the methodology part, they did not explain what the rooting theory is and how it is used and why it is used in this specific case?

Reviewer 3 Report

The topic of this research paper sounded interesting but it was one of the most difficult to read papers I have ever come across in my career. After reading several pages I would realize that I did not understand anything. It took me three days to read it and the only reason why I did not stop after the first page is because I made a commitment to MDPI to review it.

Most sentences are awkward, thus, difficult to understand. The authors have a good excuse: English is not their first language. My advice: if you are using google or another software to translate your sentences from Chinese, at least try to write shorter sentences. It works better.

The other problem is that the authors wanted the paper to sound more "scientific" so they tended to use "pompous" but empty words. Let me give one example:

"With the innovative development of new media technology, the openness of social media platforms and sharing of travel experiences are to a certain extent compatible and mutually constructive with the experience of glamping itself". WHAT?! You read paragraph after paragraph and nothing sticks. As I said, just empty words. One of the most basic things that I learned during my PhD studies (quite some time ago) is to keep it simple. Or, as one of my professors used to say "less is more". Do not use a lot jargon. The best paper is the one that can be understood even by laymen. And do not write a bunch of sentences that sound "scientific" but mean nothing. Every sentence has to mean something.

Do not get me wrong. The authors ask the right questions in the interviews and the answers given by the respondents (as well as the ones collected from different Internet platforms) sound normal. But the way the authors interpret the results is, how should I put it nicely, like commenting a different movie. Let me give just one example (but the entire paper follows the same pattern):

(lines 413-416) the authors make the following comment:

"Glamping arises from an overnight utopian experience to an ideal state of living and, moreover, an expression of freedom". My question is how do you get to associate glamping with an "utopian experience" or "an ideal state of living" or "expression of freedom"? The authors draw their conclusion based on the following comment made by a participant (XHS-9):

"Glamping alone is a romantic enjoyment; slowing down is more healing and warming"

Apart from the fact that the quote above does not sound English, how is it equivalent to what the authors concluded? In my opinion, they are not.

I was interested in the scientific methods the authors used and in the way they collected their data. This is what we are told:

"In identifying the conceptual dimensions of glamping and analyzing the process of forming a sense of atmosphere, the rooting theory approach is mainly, which helps explain the deeper patterns hidden behind the materials through a rigorous bottom-up process of utilizing inductive reasoning and gaining perspective on qualitative materials [20]" ... and then continues the same way for a few more paragraphs. So ... what did you, actually, do? If I want to, how can I replicate your study ... apart from using the "rooting theory approach" (which, by the way, needs to be explained)? You need to give me a clear recipe; fancy words do not help.

Below I will give a few more examples to illustrate my points:

line 110: Alexander Gottlieb Baugarten (sic!) (his name is actually Baumgarten) lived between 570 and 526 BC????

lines 194-195: "analyzing the building block of glamping" - What do you mean?

lines 293-294: What is an omelet chair?

line 366: What is an "organismic element"?

line 411: "becoming fodder for shared travel experiences" What do you mean?

lines 441-442: "Compared to traditional camping, glamping is more demanding of the environment, in a casual way and its symbolic meaning is more prominent." Where did the authors get this from? Was traditional camping discussed in this paper?

lines 487-491: "First, in this study, when the web data were collected due to the influence of the software's precise recommendations, the content of the notes was found to center on our geographical location by both hotness and time, so there were more notes in and around Suzhou, resulting in the findings of this paper having some regional limitations". Do you care to explain what do you mean by "hotness".

I rest my case. In conclusion, I believe the science is there but the paper needs to be carefully re-written before it can be considered for publication.

Reviewer 4 Report

The manuscript develops a qualitative analysis of the emerging phenomenon of glamping tourism. In its current state, the manuscript is well written, although at times it does not appear to have followed the authors' guidelines. The manuscript has some details that need to be addressed before it can be considered for publication. Several minor comments/changes need to be addressed before it can be considered for publication.

General comments

C1. The main concern is that the novelty of the research is not fully clear. If such novelty is not clearly highlighted, the risk is that the manuscript looks more like a simple case study rather than a research paper.

C2. In this regard, the introduction section results in some points verbose and it could be written in a more effective way, also considering the issue of cultural ecosystem services, since this is the starting point for recreation (glamping).

C3. Authors could also emphasize particular strengths of the study for potential applications of their method in other regions.

C4. For international readers involved in tourism management, what can be learned from this study?

C5. Please, look at your Discussion, are there real comparisons to other researchers of your results?. It is necessary to carry out a thorough comparison, I recommend to include some relevant references, in order to improve the discussion on the novelty of your study, compared to the others.

*The answer to these questions should be reflected in the manuscript.*

Specific comments

Line 39: Change "research. [1]" to "research [1]."

Line 48: The scope and objectives of the paper should also be clearly presented.

Line 133-136: It is important to include the concept of "scenic quality" or “scenic beauty”, as this term is, since scenic or landscape quality also defines the tourist's perception and atmosphere.

Line 149: What does "UGC" mean?

Line 169: Please add a website, add the accessed-on time as day month year.

Line 188: Why 80, how was the sample number estimated?

Line 205: Please add a website, add the accessed-on time as day month year.

Line 206: Please add a website, add the accessed-on time as day month year.

Line 230: Why not consider coordinates?

Reviewer 5 Report

Dear authors, thank you for a chance to read this interesting paper. There are just a few thigs that I would like to point to:

- The first sentence in the introduction is too long and somehow the meaning was lost; 

- since you were mentioned the stimulus-organism-response (SOR) framework, I expected to see more information about this framework in the literature review section; 

- give more information about UGC data and RED app; 

- In section 6, you mentioned for the first time Suzhou. Please give more information - did you collect information just for this region or there was some other reason for mentioning Suzhou as one of the study limitations (regional limitations)? 

- Discussion and Conclusion: The authors summarized the results obtained from the preceding qualitative analysis. However, I would like to ask whether the authors have found anything different from the previous research. It would be more interesting if the authors would indicate some differences and state the possible reasons for such differences. The inclusion of such differences would make this research much more interesting and meaningful; 

- The last section should include theoretical and practical implications, too.  

Round 2

Reviewer 2 Report

The author(s) in the new version made some improvements. However, In the theoretical section, the cite given to Lefebvre based on perceived, conceived and lived space is not connected in the elaboration of glamping experience not only in methodology but also in the findings and conclusion part. This should be connected otherwise, there is no meaning of citing this important theoretical elaboration.

Reviewer 3 Report

It reads much better this time. Thank you for taking the time to improve sentence readability 

Author Response

Dear professor:

I am very happy that the revised manuscript is now somewhat readable. The quality of the manuscript has been improved because of your valuable comments. Thank you again for your careful and responsible review of the manuscript, and we hope that the paper will be published successfully in the near future.

Kind regards.

Reviewer 5 Report

The authors provided much improved version of the article and paid good attention to reviewer comments.

Author Response

Response

Dear professor:

I am very happy that the revised manuscript is now somewhat readable. The quality of the manuscript has been improved because of your valuable comments. Thank you again for your careful and responsible review of the manuscript, and we hope that the paper will be published successfully in the near future.

Kind regards.

Round 3

Reviewer 2 Report

I have seen that the author(s) made some improvements to the criticisms.